# Ynimines as versatile precursors to 2-imido- and 2-amido-1,3-dienes for stereodivergent diels–alder reactions

Ruijia Wang[1,2], Xin-Qi Zhu[2,3], Maël Djaïd[2], Rémi Lavernhe[2], Qian Wang[2] & Jieping Zhu[2] ✉

In contrast to ynamides, whose chemistry has been extensively explored, ynimines remain underutilized in organic synthesis despite their rich functionalities. Here we report a general strategy to access 2-imido-1,3-dienes, synthetically challenging building blocks, through the reaction of ynimines with carboxylic acids. Leveraging this transformation, we develop a three-component reaction of ynimines, carboxylic acids and electron-deficient alkenes that enables the efficient synthesis of 1-imido-3,4-*trans*-disubstituted cyclohex-1-enes. The sequence proceeds via regioselective hydroacyloxylation and Mumm rearrangement to generate 2-imido-1,3-dienes, which undergo Diels–Alder cycloadditions. An intramolecular variant furnishes *trans*-fused tricyclic architectures reminiscent of *trans*-Δ⁹-tetrahydrocannabinol. Chemoselective hydrolysis further converts 2-imido-1,3-dienes into 2-amido-1,3-dienes, enabling chiral squaramide-catalysed enantioselective Diels–Alder reactions to afford 1-amido-3,4-*cis*-disubstituted cyclohex-1-enes with high stereocontrol. Distinct concerted and stepwise cycloaddition pathways rationalize the observed stereodivergence.

Two types of (acyl)amino substituted 1,3-dienes are known, with Rawal's 1-dimethylamino-3-silyoxy-1,3-diene being a notable example (Fig. 1a)[1–4]. Interestingly, while the cycloaddition reactions of 1-amido-1,3-dienes 1[5–19], including catalytic enantioselective variants, have been well documented for the synthesis of allylic amines[20,21], research on the 2-amido counterpart 2 has largely lagged behind. This is unfortunate, as the cycloaddition of 2-amido-1,3-dienes 2 affords an enamide function which is poised for further functional group transformations[22–24]. The lack of progress in this area is likely due to the absence of a convenient and general synthetic method for preparing 2. In contrast, the chemistry of 2-amino-1,3-dienes has been comparatively more explored[25–27].

We have recently reported the examples of chiral phosphoric acid-catalyzed enantioselective Diels-Alder reaction of 2-trifluoroacetamido-1,3-dienes 2 with electron-deficient dienophiles 3 (Fig. 1b)[28] as well as higher order cycloaddition between 2 and tropones[29]. The desired 1,3-dienes 2 was prepared by a reductive acylation of α,β-unsaturated oximes with excess of iron powder and trifluoroacetic anhydride, a protocol originally developed for enamide synthesis[30–32]. However, the use of single electron transfer reducing agent (Fe) in combination with a strongly electrophilic reagent [$(RCO)_2O$] limited functional group tolerance. We have also explored a one-pot $Ti(OiPr)_4$-promoted condensation of ammonia with α,β-unsaturated ketones followed by N-acylation of the resulting imine with acetic anhydride[33]. However, both 1-acetamido and 2-acetamido-1,3-dienes were formed in the cases of β-alkyl substituted α,β-unsaturated ketones in line with the literature precedents[34].

[1]School of Pharmacy, Hunan University of Chinese Medicine, Changsha, China. [2]Laboratory of Synthesis and Natural Products (LSPN), Institute of Chemical Sciences and Engineering Ecole Polytechnique Fédérale de Lausanne, EPFL-SB-ISIC-LSPN, BCH 5304, 1015 Lausanne, Switzerland. [3]Yunnan Key Laboratory of Modern Separation Analysis and Substance Transformation, College of Chemistry and Chemical Engineering, Yunnan Normal University, Kunming, China. ✉e-mail: jieping.zhu@epfl.ch

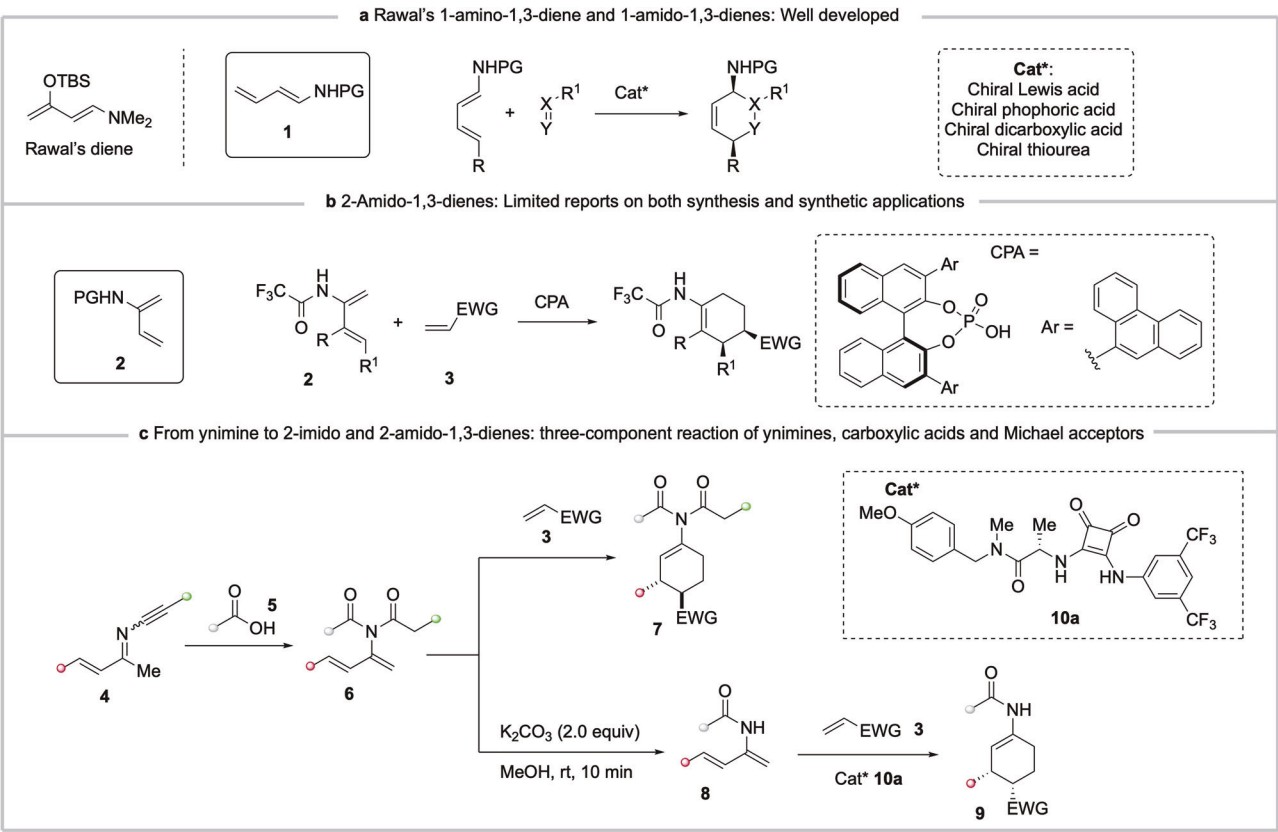

**Fig. 1 | Literature precedents and this work. a**, Rawal's 1-amino-1,3-diene and 1-amido-1,3-dienes: Well developed. **b**, 2-Amido-1,3-dienes: Limited reports on both synthesis and synthetic applications. **c**, From ynimine to 2-imido and 2-amido-1,3-dienes: three-component reaction of ynimines, carboxylic acids and Michael acceptors. EWG= electron withdrawing group. CPA = chiral phosphoric acid.

To fully exploit the chemistry of 2-acetamido-1,3-dienes **2**, an alternative and more general synthesis under mild conditions is required. We considered ynimines **4**[35–39], now easily accessible[40–42], as suitable precursors. Herein, we report that the reaction of ynimines with carboxylic acids provides a convenient access to 2-imido-1,3-dienes **6** via a regioselective hydroacyloxylation of the alkyne function in **4**, followed by a Mumm rearrangement of the resulting enol esters. A three-component reaction of ynimines **4**, carboxylic acids **5** and α,β-unsaturated carbonyl compounds **3** is subsequently developed affording 1-imido-*trans*-3,4-disubstituted cyclohex-1-enes **7** in good yields with excellent diastereoselectivity (Fig. 1c). Importantly, selective hydrolysis of one of the acyl groups in compounds **6** furnishes the 2-amido-1,3-dienes **8**, which, in the presence of a catalytic amount of chiral squaramide **10a**, undergo enantioselective Diels-Alder reaction with dienophiles **3** to give 1-amido-*cis*-2,4-disubstituted cyclohex-1-enes **9** in good yields with high enantioselectivity. The observed stereodivergence is attributed to a concerted *exo* cycloaddition of 2-imido-1,3-dienes, as opposed to a stepwise mechanism operative for the more nucleophilic 2-amido-1,3-dienes.

## Results

**From Ynimines to 2-imido-1,3-dienes and 2-amido-1,3-dienes:** Three-component synthesis of 1-imido-*trans*-3,4-disubstituted cyclohex-1-enes. Both palladium-catalyzed (toluene, 70 °C)[43] and metal-free hydroacyloxylation (toluene, 100 °C) of ynamides to give α-acyloxyenamides are known[44–50]. However, the multiple functional nature of ynimines raised concerns about competing side reactions, but also offered the opportunity to develop domino process inaccessible to ynamides. Gratifyingly, reaction of ynimine **4a** with benzoic acid **5a** at room temperature furnished the α-alkylidene imino enol

ester **11a** in 95% yield. The fact that this reaction occurred at room temperature is consistent with the higher reactivity of ynimines compared to ynamides. Heating **11a** in DCE at reflux promoted a smooth Mumm rearrangement to deliver 2-imido-1,3-diene **6a** in 90% yield, with no evidence for the competing 6π-electrocyclization to dihydropyridine **12** (Fig. 2a, inset)[51]. Finally, the Diels-Alder reaction of **6a** with methyl vinyl ketone (MVK, **3a**) in refluxing DCE provided the expected cycloadduct **7a** in 83% isolated yield. To the best of knowledge, 2-imido-1,3-dienes have scarcely been reported[32], and their Diels-Alder cycloaddition reactions remain unexplored.

Interestingly, the two acyl groups of the imido diene **6a** can be selectively removed to access two distinct 2-amido-1,3-dienes. Thus, treatment of **6a** with potassium carbonate ($K_2CO_3$) and methanol in DCE effected selective methanolysis of the phenylacetyl group, furnishing benzamide **8a** (83% yield), whose structure was confirmed by X-ray crystallographic analysis. In contrast, reaction of **6a** with pyrrolidine promoted selective aminolysis of the benzoic group, affording in **13a** in 81% yield along with *N*-benzoyl pyrrolidine (80%).

The imido-1,3-diene **6a** was stable and can be purified by conventional methods. The fact that no external reagent was required to promote the reaction of **4a**, **5a** and **3a** prompted us to develop a one-pot three-component synthesis of **7a** (Fig. 2b). Gratifyingly, stirring a DCE solution of the three reactants at room temperature until the complete consumption of **4a** followed by heating to reflux the DCE solution afforded the 1-imido-*trans* 3,4-disubstituted cyclohex-1-ene **7a** in 66% isolated yield. We have also briefly assessed the solvent effect. Whereas toluene was an efficient solvent affording **7a** in a slight lower yield (60%) than DCE, others such as dioxane, MeCN, DMF, PhCF₃ and PhCl were far less good reaction media. Notably, six chemical bonds (one $C_{sp^2}$-O, one $C_{sp^2}$-N, two $C_{sp^3}$-$C_{sp^3}$ and two $C_{sp^3}$-H) were formed in

this operationally simple procedure. No external reagents, except for heating, is needed to drive this three-component reaction to completion[52].

The scope of the three-component reaction was next examined (Fig. 3). The reaction tolerated a wide range of aryl groups at the β position of the conjugated ketimine motif. Both electron donating groups (OMe, Me) and electron-withdrawing groups (F, Cl, Br) at the *para*, *meta*, and *ortho* positions were well tolerated under the reaction conditions (**7a-7i**). Ynimines bearing a disubstituted phenyl (**7j, 7k**), a naphthyl (**7l**), or a thiophene (**7m**) also proved to be competent substrates. However, the ynimine bearing a 4-nitrophenyl group at its β position (Ar = 4-NO₂C₆H₄) led to an unstable cycloadduct that decomposed upon purification. Other dienophiles, such as ethyl vinyl ketone (EVK, **7n**), ethyl acrylate (**7o**), 4-bromophenyl acrylate (**7p**) and 4-chlorophenyl acrylate (**7q**), successfully participated in this three-component reaction. However, no cycloaddition products were observed when acrylonitrile and nitroethylene were used as dienophiles, likely due to the facile polymerization of these highly active species. Finally, glycolic acid participated in this three-component reaction to afford cycloadduct **7r** in 55% yield with diminished diastereoselectivity (dr 5:1).

The reaction of β-alkyl substituted ynimines with carboxylic acids was next examined. Not surprisingly, the reaction of **4s** with benzoic acid (**5a**) under refluxing in DCE afforded a roughly 1:1 mixture of two dienes: 2-imido-1,3-diene **6s** and 1-imido-1,3-diene **14s** (Fig. 4a). We note that no equilibrium was observed when either **6s** or **14s** was re-submitted to the reaction conditions (DCE, 100 °C).

To favour the formation of **6s**, various bases were screened and 2,2,6,6-tetramethylpiperidine (TMP), was identified as optimal (See Supplementary Information, page S18, Table S4). Experimentally, heating a DCE solution of **4s, 5a** and TMP (1.3 equiv) furnished **6s** and **14s** in yields of 83% and 8%, respectively. Finally, the three-component reaction of **4s, 5a**, and phenyl vinyl ketone (**3f**) in the presence of TMP afforded directly the 3,4-disubstituted 1-imidocyclohex-1-ene **7s** in 73% isolated yield (Fig. 4b).

As shown in Fig. 4b, a variety of dienophiles, including 4-methoxy, 4-chloro, 4-bromo and 4-cyanophenyl and thiophenyl vinyl ketones, readily participated in this reaction to afford the three-component adducts (**7t-7x**) in good yields with excellent diastereoselectivity. Moreover, ynimines containing functional groups such as silyl ether and cyano substituents also proved to be competent substrates, delivering the corresponding cyclohexene derivatives **7y** and **7z**.

The 3,4-*trans* relative stereochemistry of **7k** (Fig. 3) was unambiguously determined by X-ray crystallographic analysis. For all compounds depicted in Fig. 3 and Fig. 4, the coupling constants $J_{H3-H4}$ and $J_{H2-H3}$ were found to be in the range of 8.4-8.9 Hz and 2.3-3.2 Hz, respectively. Accordingly, a 3,4-*trans* stereochemistry was assigned for all these cyclohexenes.

## Organocatalytic enantioselective Diels-Alder reaction of 2-amido-1,3-dienes

We previously reported a chiral phosphoric acid-catalyzed enantioselctive cycloaddition of 2-trifluoroacetamido-1,3-dienes with electron-deficient dienophiles[28]. However, these conditions proved unsuitable for other 2-acylamino-1,3-dienes due to the decomposition of these more nucleophilic dienes. With ready access to 2-amido-1,3-dienes from ynimines, we thought to develop a more general catalytic enantioselective protocol for this family of 1,3-dienes. Using ynimine **4a** (R = Ph) as a model substrate, its catalytic enantioselective transformation was investigated as follows. A solution of ynimine **4a** (R = Ph) and benzoic acid **5a** (R¹ = Ph) in DCE was heated to reflux for 15 h, then cooled to room temperature, diluted with MeOH, and treated with K₂CO₃. After aqueous work-up, the crude 2-amido-1,3-diene (**8a**, R = R¹ = Ph) was directly subjected to a D-A reaction with 4-chlorophenyl vinyl ketone (**3h**) at room temperature in the presence of organocatalysts (Fig. 5). Consistent with our previous report, decomposition of **8a** was observed in the presence of chiral phosphoric acids (CPA)[53–56] and *N*-triflyl phosphoramides[57].

We therefore turned our attention to other H-bonding organocatalysis (See Supplementary Information, page S18-20, Tables S5 and S6). Screening sixteen chiral squaramides[58–62], three

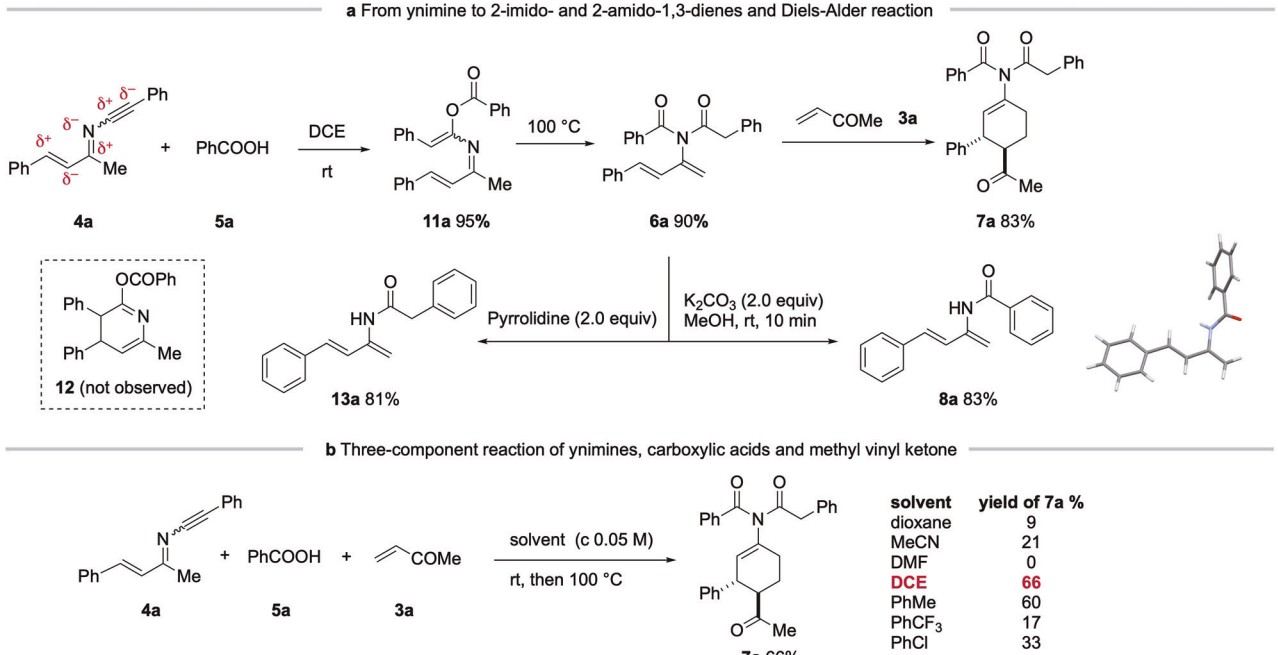

**Fig. 2 | Synthesis of 2-imido-1,3-dienes and 2-amido-1,3-dienes from ynimines: three-component reaction of ynimine 4a, benzoic acid 5a and methyl vinyl ketone 3a for the synthesis of 1-imido-3,4-*trans*-disubstituted cyclohex-1-enes.** **a**, From ynimine to 2-imido- and 2-amido-1,3-dienes and Diels-Alder reaction. **b**, Three-component reaction of ynimines, carboxylic acids and methyl vinyl ketone. Yields refer to the isolated pure products. DCE = dichloroethane.

**Fig. 3 | Scope of the three-component reaction of ynimines, carboxylic acids and α,β-unsaturated carbonyl compounds.** Reaction conditions: **4** (0.1 mmol), **5** (1.3 equiv), **3** (2.0 equiv), DCE (2.0 mL), rt, then 100 °C. Yields refer to the isolated pure products.

thioureas[63–65] and two ureas bearing a single H-bond donor[66] revealed squaramide **10a** as the most effective catalyst[67,68], affording the cycloadduct **9a** with 67% *ee*. Gratifyingly, conducting the same reaction at -20 °C furnished **9a** in 48% isolated yield with 93% *ee*.

The generality of this catalytic enantioselective cycloaddition of 2-amido-1,3-dienes, generated from ynimines without purification, was next explored, with particular attention being paid on the structural diversity of the carboxylic acids. As shown in Fig. 5, benzoic acids bearing either electron-donating or electron-withdrawing substituents were well tolerated, affording adducts **9a-9f**. Both 2-naphthoic acid and biphenyl-4-benzoic acid also participated in the reaction, delivering the corresponding adducts **9 g** and **9 h**, respectively, in good yields with high *ees*. An ynimine bearing a β-alkyl substituent was converted to cyclohexene **9i** in 43% yield with 88% ee. In addition to α,β-unsaturated ketones, methyl acrylate underwent cycloaddition but provided the cycloadduct with low *ee*. The absolute and relative stereochemistry of compound **9 h** was determined by X-ray crystallographic analysis. For all compounds listed in Fig. 5, the coupling constants $J_{H3-H4}$ and $J_{H2-H3}$ are found in the ranges of 5.1-5.6 Hz and 4.0-5.4 Hz, respectively, supporting the assignment of a 3,4-*cis* stereochemistry for all cyclohexenes.

### Intramolecular Diels-Alder reaction of in situ generated 2-imido-1,3-dienes

The observation that the presence of dienophiles did not interfere with the generation of 2-imido-1,3-dienes from the reaction of ynimines with carboxylic acid prompted us to explore an intramolecular variant of the D-A cycloaddition reaction. Ynimine **15a** (R = Cl) was accordingly

selected as a model substrate for this study (Fig. 6). Gratifyingly, heating a DCE solution of **15a** (R = Cl) and benzoic acid (**5a**) at 100 °C furnished the *trans*-fused tricyclic compound **16a** in 45% isolated yield. The reaction is proposed to proceed through the imido-diene intermediate **17**, which undergoes an intramolecular D-A reaction to deliver the observed cycloadduct. Analogous compounds **16b-16d** were obtained in comparable yields. The structure of **16b** was confirmed by X-ray crystallographic analysis. Although the overall yield appears to be moderate, it should be noted that the conversion of **15** and **5a** to **16** involves at least five individual bond-forming events, including the formation of two C-C, one C-N, one C = O, and two C-H bonds. On this basis, the effective yield per bond formation is comparatively high.

The presence of aryl halide functions in compounds **16a-16b** provides useful handles for further diversification through transition metal catalyzed cross-coupling reactions. Notably, compounds **16** can be viewed as structural analogues of *trans* Δ[9]-tetrahydrocannabinol **18** (Fig. 6 inset), the principal psychoactive constituent of marijuana and an FDA-approved drug (dronabinol) for the treatment of HIV/AIDS-induced anoxia[69] and chemotherapy-induced nausea and vomiting[70].

### Stereochemical outcomes

The stereodivergence observed in the cycloaddition of dienophile **3** with 2-imido-1,3-dienes **6** versus 2-amido-1,3-dienes **8** is of synthetic significance and warrants further discussion. Control experiments showed that the 3,4-*trans* diastereomer **7a** was formed in the reaction of **6a** with **3a**, regardless of the reaction temperature (20 °C to 100 °C). Monitoring the reaction progress under standard conditions revealed that **7a** remained the sole cycloadduct at different conversion levels. In

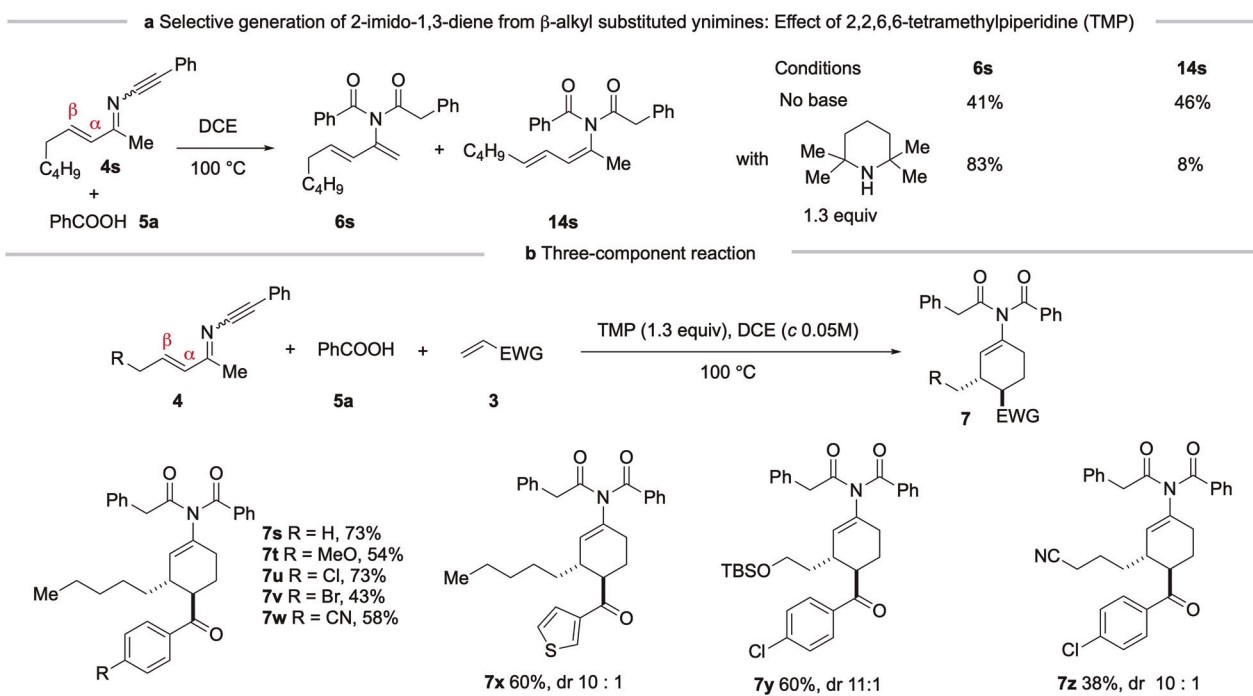

**Fig. 4 | Selective generation of 2-imido-1,3-dienes from β-alkyl substituted ynimines and scope of the three-component reaction. a**, Condition optimization; **b**, Scope. Reaction conditions: **4** (0.1 mmol), **5a** (1.3 equiv), TMP (1.3 equiv), 3 (3.0 equiv), DCE (2.0 mL), RT, then 100 °C. Yields refer to the isolated pure products. TMP = 2,2,6,6-tetramethylpiperidine.

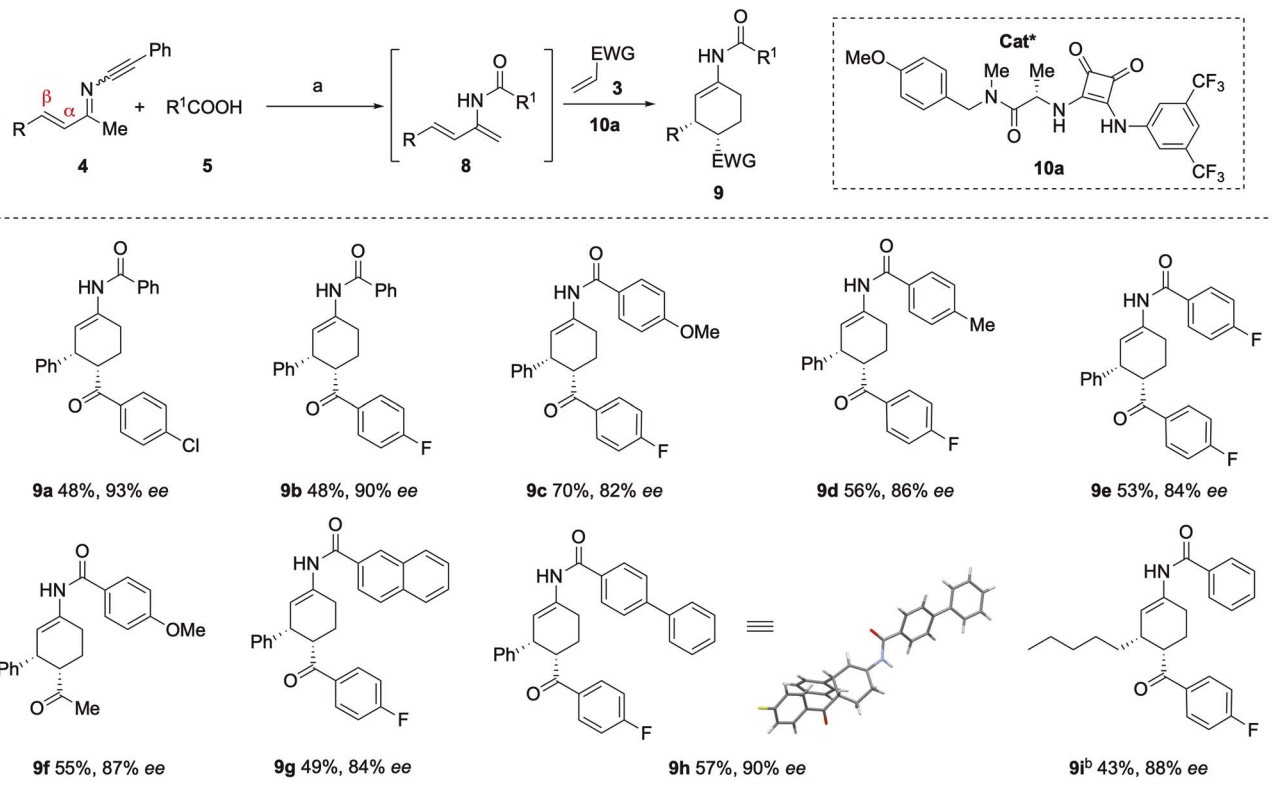

**Fig. 5 | Squaramide-catalyzed enantioselective cycloaddition of 2-amido-1,3-dienes with α,β-unsaturated ketones.** Conditions a: **4** (0.1 mmol), **5** (1.3 equiv), DCE (2.0 mL), 100 °C, 15 h, then K₂CO₃ (2.0 equiv), methanol (2.0 mL), rt, 10 min. After usual workup, the crude product **8**, α,β-unsaturated ketones **3** (2.0 equiv), squaramide **10a** (0.2 equiv), DCE (2.0 mL), -20 °C; Conditions. b: 2,2,6,6-tetramethylpiperidine (1.3 equiv) was added. Yields refer to the isolated pure products.

**Fig. 6 | Intramolecular trapping of the in situ generated 2-imido-1,3.dienes: Synthesis of trans-fused tricyclic compounds reminiscent of trans Δ9-tetra-hydrocannabinol.** Reaction conditions: **15** (0.1 mmol), PhCOOH (1.3 equiv), DCE (2.0 mL), 100 °C. Yields refer to the isolated pure products.

**Fig. 7 | Cis-selective cycloaddition of 2-amido-1,3-diene. Conditions a: 8a (0.1 mmol), 3 h (2.0 equiv), DCE (2.0 mL), 20 °C. Conditions b: 8a (0.1 mmol), 3 h (2.0 equiv),** squaramide **10a** (0.2 equiv), DCE (2.0 mL), -20 °C.

contrast, the cycloaddition of 2-amido-1,3-diene **8a** with 4-chlorophenyl vinyl ketone (**3 h**) proceeded readily at room temperature to afford two diastereomers, the *cis*-**9a** and *trans*-**9a** in 85% yield with a dr of 5:1 (Fig. 7). No epimerization was observed when a solution of diastereomerically pure *cis*-**9a** or *trans*-**9a** was heated to reflux for 5 h.

The results of these control experiments suggested that a) both *trans*-**7** and *cis*-**9**, obtained from the cycloaddition of imidodiene **6** and amidodiene **8**, respectively, are kinetic products, and no epimerization took place under reaction conditions; b) amidodiene **8** is significantly more reactive than imidodiene **6** towards electron-poor dienophiles, likely because the nitrogen in **6** bears two acyl groups, reducing thereby the nucleophilicity of the enamide unit; c) the presence of chiral catalyst **10a** significantly enhanced the diastereoselectivity of the cycloaddition of **8**. In the absence of the catalyst, the racemic cyclohexene **9a** was formed as a mixture of *cis/trans* diastereomers in 5 to 1 ratio, whereas in the presence of chiral squaramide **10a**, the enantioenriched **9a** was formed with a dr > 20:1 (Fig. 7).

Based on the aforementioned results, we hypothesized that the Diels-Ader reaction of **6** with **3** proceeded via a concerted mechanism, with the *exo*-TS favoured over the alternative *endo*-TS. In the *endo*-TS,

any stabilizing secondary orbital interaction between the carbonyl group and the π* of diene is offset by the sever steric repulsion. In addition, the 2-imido function in the *exo*-TS may engage with the π* of the diene, lowering thereby the activation energy. Moreover, the 3,4-*trans*-**7** was expected to be thermodynamically more stable than the *cis*-isomer, as the latter suffered from the steric clash between the C4 substituent and one of the N-acyl groups (Fig. 8a).

For the intramolecular D-A reaction of imido-diene **17**, generated in situ from ynimine **15**, the *exo*-TS would also be favoured for the same reasons as in the intermolecular version (Fig. 8b). Interestingly, the benzo-tethered, ester-linked 1,3,9-decatriene also afforded the *trans*-fused tricycle as a major product, albeit with lower *trans/cis* selectivity (dr 3:1)[71]. Therefore, the presence of a 2-imido function in the 1,3-diene system of **18** improved significantly the diastereoselectivity of the cycloaddition reaction.

Finally, we propose a stepwise mechanism for the squaramide **10a**-catalyzed enantioselective cycloaddition of 2-amido-1,3-dienes **8** with α,β-unsaturated carbonyl compounds **3**. Indeed, most cycloadditions involving enamides are known to proceed through a stepwise pathway[22–24,72–80]. The reaction likely began with a Michael addition of the enamide moiety of **8** to **3** in the presence of **10a**, generating

**Fig. 8 | Stereochemical divergence in the cycloaddition of 2-imido- and 2-amido-1,3-dienes with electron-deficient dienophiles: Concerted mechanism vs stepwise pathway. a** Concerted *exo*-selective cycloaddition between 2-imido-1,3-dienes **6** and electron-poor dienophiles **3**. **b** Concerted *exo*-selective intramolecular cycloaddition of **17**. **c** Stepwise reaction pathway in squaramide-catalyzed reaction between 2-amido-1,3-dienes and electron-poor dienophiles **3**.

intermediate **19**. This is followed by an intramolecular 1,4-addition of the resulting enolate to the α,β-unsaturated *N*-acyliminium ion via a synclinal TS, yielding the *cis*-adduct **9**. In this TS, both the nucleophile and electrophile are activated through multiple hydrogen bonding interactions with the squaramide catalyst, lowering the activation barrier for C-C bond formation. Notably, no reaction occurred at -20 °C without catalyst **10a**, underscoring its crucial role in both activating the reactants and facilitating the enantio-determining C-C bond forming step in the proposed stepwise pathway.

## Discussion

In conclusion, we report a synthesis of 2-imido-1,3-dienes, a class of building blocks that is otherwise difficult to access, via the reaction of ynimines with carboxylic acid. Building on this transformation, we developed a three-component reaction of ynimines, carboxylic acids, and electron-deficient alkenes for the synthesis of 1-imido-3,4-*trans*-disubstituted cyclohex-1-enes. The transformation proceeds through a regioselective hydroacyloxylation of the ynimine, followed by a Mumm rearrangement and a Diels-Alder cycloaddition. An intramolecular variant enables access to *trans*-fused tricyclic scaffold, reminiscent of *trans*-Δ⁹-tetrahydrocannabinol. Moreover, chemoselective hydrolysis of 2-imido-1,3-dienes generates 2-amido-1,3-dienes, which undergo chiral squaramide-catalyzed enantioselective Diels-Alder reactions to furnish 1-amido-3,4-*cis*-disubstituted cyclohex-1-enes with excellent diastereo- and enantioselectivity. The stereochemical divergence is rationalized by a concerted *exo*-TS for 2-imido-1,3-dienes and a stepwise pathway for 2-amido-1,3-dienes, highlighting the interplay of steric and electronic effects in these transformations.

## Methods

**General procedure for the three-component reaction of ynimines 4, carboxylic acids 5 and dienophiles 3 for the synthesis of 1-imido-3,4-*trans*-disubtuituted cyclohexe-1-enes 7**
A mixture of ynimine **4** (0.10 mmol), carboxylic acids **5** (1.3 equiv) and dienophile **3** (2.0 equiv) in DCE (2.0 mL, 0.05 M) was stirred at room temperature until complete consumption of **4**. The reaction mixture was then heated to reflux until the in situ generated 2-imido-1,3-diene was fully consumed. The reaction was then quenched with water and the aqueous phase was extracted with dichloromethane. The combined organic phases were washed with brine, dried over MgSO₄, filtered, and concentrated under vacuum. The residue was purified by flash chromatography on silica gel to provide the desired product **7**.

## Data availability

Crystallographic data for the structure reported in this article have been deposited at the Cambridge Crystallographic Data Centre, under deposition numbers CCDC 2260903 (**7k**), CCDC 2324899 (**8a**), CCDC 2324850 (**9 h**) and CCDC 2324851 (**16b**). Copies of the data can be obtained free of charge via https://www.ccdc.cam.ac.uk/structures/. The data supporting the findings of this study are available within the article and its Supplementary Information files. All data is available from the corresponding author upon request.

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

## Acknowledgements

We thank EPFL (Switzerland) and Swiss National Science Foundation (SNSF 20021-219764) for financial supports. RW thanks China Scholarship Council for fellowship (201908430132) and Scientific Research Fund of Hunan Provincial Education Department (N° 21A0245). XQZ thanks Yunan Normal University and Xiamen University for postdoctoral fellowships. We thank Dr. F. Fadaei-Tirani and Dr. R. Scopelliti for the X-ray structural analysis of compounds **7k**, **8a**, **9h**, and **16b**.

## Author contributions

R.W., X.Q.Z.,Q.W., J.Z. conceived and designed the experiments. R.W., X.Q.Z., M.D., R. L. carried out the experiments. R.W., Q.W., J.Z. interpreted the results and co-wrote the manuscript.

## Competing interests

The authors declare no competing interests.

## Additional information

**Peer review information** . *Nature Communications* thanks the anonymous reviewer(s) for their contribution to the peer review of this work. A peer review file is available.

