## [Transparent Peer Review file · Nature Communications]

Ynimines as Versatile Precursors to 2-Imido- and 2-Amido-1,3-Dienes for Stereodivergent Diels–Alder Reactions

Corresponding Author: Professor Jieping Zhu

Version 0:

Reviewer comments:

Reviewer #1

(Remarks to the Author)

This paper describes stereoselective Diels-Alder reactions with ynimines, as precursors of 2-imido- and 2-amido-1,3-dienes. The authors conducted many reactions and are actively investigating them; however, this reviewer has several comments as mentioned below.

1. Since the chemical yield of this reaction is low and the ee is not very good, it is not very attractive from the perspective of organic synthetic chemistry.
 2. Similar structures to 2-amido-1,3-dienes **6a** are commercially available. This reviewer thinks the 1-imido form also can be derived from this, and this reaction can be achieved without **4a**, which must be synthesized separately. Therefore, this synthetic method is not particularly attractive.
 3. The asymmetric catalyst is not a new one that was newly developed by the authors.
 4. More detailed consideration of the reaction mechanism using calculations, etc. is needed.
 5. There is no mention of applications of the products (natural products, biologically active substances, etc.).
- Thus, given the criteria of *Nature Communication*, this reviewer does not think this paper meet the criteria, and cannot accept it. This reviewer recommends submitting to another journal.

Reviewer #2

(Remarks to the Author)

Comments: 1-Amido-cyclohex-1-ene motifs are widely found in the pharmaceutical, natural products and biologically active compounds. This manuscript by Zhu and coworkers reported a facile method for the synthesis of 2-imido-1,3-dienes via the reaction of ynimines with carboxylic acid. Authors found that a three-component reaction of ynimines, carboxylic acids and electron-deficient alkenes enable the efficient synthesis of 1-imido-3,4-trans disubstituted cyclohex-1-enes. Furthermore, the 2-imido diene can be selectively removed to access two distinct 2-amido-1,3-dienes, which undergo chiral squar amide-catalyzed enantioselective Diels-Alder reactions to furnish 1-amido-3,4-cis-disubstituted cyclohex-1-enes with excellent diastereo- and enantioselectivity. The stereochemical divergence is proposed by a concerted exo-TS for 2-imido-1,3-dienes and a stepwise pathway for 2-amido-1,3-dienes. In my opinion, publication of the manuscript may be recommended in Nature Communications after a few points were supplemented and revised, which are as follows:

1. In Page 3 line 24, the Fig. 2c should be Fig. 2b.
2. In Fig 3, author investigate different dienophiles partners, such as ethyl vinyl ketone, ethyl acrylate, 4-bromo phenyl acrylate, and 4-chlorophenyl acrylate, I think that acrylonitrile and nitroethylene should be added.
3. As for Fig 3, only **7i** with dr 11:1, other 7 products were all single trans-figure? If yes, I think that author should mention it in the manuscript.
4. In three-component reaction, can the alkyl acid afford the desired products?
5. Many formatting errors need to be corrected, such as reference 20, 24, 27, 30, 37, etc. letter of all words should be capital.
6. Some NMR spectra are not clean, including impurities or residual solvents (1H NMR-4l, 6a). please provide clean copies.

Reviewer #3

(Remarks to the Author)

Zhu et al. present a new method for synthesizing 2-imido-1,3-dienes from ynimines and carboxylic acids. They further introduce a three-component reaction with ynimines, carboxylic acids, and electron-deficient alkenes to create 1-imido-3,4-trans-disubstituted cyclohex-1-enes via hydroacyloxylation, Mumm rearrangement, and Diels–Alder cycloaddition. An intramolecular version accesses trans-fused tricyclic scaffolds similar to trans- Δ^9 -tetrahydrocannabinol, demonstrating this strategy's versatility. Chemoselective hydrolysis yields 2-amido-1,3-dienes, which undergo chiral squaramide-catalyzed Diels–Alder reactions for 1-amido-3,4-cis-disubstituted cyclohex-1-enes with high selectivity. The different stereochemical outcomes arise from distinct mechanistic pathways influenced by sterics and electronics. Overall, this work offers strong atom economy and broad utility and is recommended for publication in Nature Communications after minor revisions noted below.

Questions:

- Were strongly electron-withdrawing β -aryl groups (e.g., nitro, cyano, CF₃) tested?
- For substrate 7i in Figure 3, is there a mixture of regioisomers/diastereomers, and which predominates? How was selectivity determined? (e.g. based on 1H NMR or GC/HPLC of crude reaction mixtures?)
- Did the study assess 2-imido-1,3-dienes, acrylates, or arylonitriles as dienophiles in enantioselective Diels–Alder reactions?
- Can the authors clarify if side products formed or if starting material was recovered, given lower yields in both the enantioselective and the intramolecular Diels–Alder reactions?
- Was telescoping (two consecutive Diels–Alder reactions using ortho-hydroxybenzhydryl alcohols) attempted as in prior work [Angew. Chem. Int. Ed. 62, e202214925 (2023)]?

Corrections:

- In Figure 3, replace RCOOH with PhCO₂H since only benzoic acid was used.
- The experimental molecular formula for compound 7s does not match its reported HRMS data.

Version 2:

Reviewer comments:

Reviewer #2

(Remarks to the Author)

All raised issues have been thoroughly revised and supplemented. This revised manuscript meets your journal's standards, and we respectfully request it be considered for publication.

Reviewer #3

(Remarks to the Author)

The authors have satisfactorily addressed all the reviewer comments and incorporated the suggested changes in the revised manuscript. The manuscript is suitable for publication in Nature Communications in its current form.

Lausanne, 07-January-2026

Dear Editor

Please find enclosed our revised manuscript entitled “Ynimines as Versatile Precursors to 2-Imido- and 2-Amido-1,3-Dienes for Stereodivergent Diels–Alder Reactions” (MS N° NCOMMS-25-96860). We thank reviewers for their thoughtful comments. The manuscript has been revised following your advice and reviewers’ suggestions.

Reviewer #1

1. *Since the chemical yield of this reaction is low and the ee is not very good, it is not very attractive from the perspective of organic synthetic chemistry.*

2. *Similar structures to 2-amido-1,3-dienes 6a are commercially available. This reviewer thinks the 1-imido form also can be derived from this, and this reaction can be achieved without 4a, which must be synthesized separately. Therefore, this synthetic method is not particularly attractive.*

3. *The asymmetric catalyst is not a new one that was newly developed by the authors.*

4. *More detailed consideration of the reaction mechanism using calculations, etc. is needed.*

5. *There is no mention of applications of the products (natural products, biologically active substances, etc.).*

We respectfully note that we do not share the reviewer’s assessment. Furthermore, the comments are formulated in fairly broad terms, which makes it difficult for us to respond to them point-by-point.

Reviewer #2

“1-Amido-cyclohex-1-ene motifs are widely found in the pharmaceutical, natural products and biologically active compounds. This manuscript by Zhu and coworkers reported a facile method for the synthesis of 2-imido-1,3-dienes via the reaction of ynimines with carboxylic acid. Authors found that a three-component reaction of ynimines, carboxylic acids and electron-deficient alkenes enable the efficient synthesis of 1-imido-3,4-trans disubstituted cyclohex-1-enes. Furthermore, the 2-imido diene can be selectively removed to access two distinct 2-amido-1,3-dienes, which undergo chiral squaramide-catalyzed enantioselective Diels-Alder reactions to furnish 1-amido-3,4-cis-disubstituted cyclohex-1-enes with excellent diastereo- and enantioselectivity. The stereochemical divergence is proposed by a concerted exo-TS for 2-imido-1,3-dienes and a stepwise pathway for 2-amido-1,3-dienes. In my opinion, publication of the manuscript may be recommended in Nature Communications after a few points were supplemented and revised, which are as follows”

We thank the reviewer for the positive and encouraging comment and have addressed the points raised below.

Regarding “1. In Page 3 line 24, the Fig. 2c should be Fig. 2b.” Corrected.

Regarding 2. *in Fig 3, author investigate different dienophiles partners, such as ethyl vinyl ketone, ethyl acrylate, 4-bromo phenyl acrylate, and 4-chlorophenyl acrylate, I think that acrylonitrile and nitroethylene should be added.*” We have examined two dienophiles. However, no cycloaddition products were observed, likely due to the facile polymerization of these highly active species. A sentence was added and it reads (page 4, lines 18-19): “*However, no cycloaddition products were observed when acrylonitrile and nitroethylene were used as dienophiles, likely due to the facile polymerization of these highly active species.*”

Regarding “3. *As for Fig 3, only 7i with dr 11:1, other 7 products were all single trans-figure? If yes, I think that author should mention it in the manuscript.*” This was a typo. Only one diastereomer was detected for **7i**, consistent with the other compounds of this series. This has been corrected in the revised version.

Regarding “4. *in three-component reaction, can the alky acid afford the desired products?*” Glycolic acid was used and the reaction afforded the cycloadduct in good yield but with reduced diastereoselectivity. A sentence was added and it reads (page 4, line 19, and page 5, lines 1-2): “*Finally, glycolic acid participated in this three-component reaction to afford cycloadduct **7r** in 55% yield with diminished diastereoselectivity (dr 5:1).*”

Regarding “5. *Many formatting errors need to be corrected, such as reference 20, 24, 27, 30, 37, etc. letter of all words should be capital.*” It has been corrected in the revised version. All other references have been verified carefully and the first letter of each word in the title are capitalized.

Regarding “6. *Some NMR spectra are not clean, including impurities or residual solvents (1H NMR-4l, 6a). please provide clean copies.*” The ¹H and ¹³C NMR spectra of compound **6a** recorded in CD₃CN exhibit significantly higher resolution than those previously obtained in CD₃Cl. Accordingly, the earlier data (Page S17) and spectra (Page S96-S97) have been replaced with those acquired in CD₃CN. Isolation of compound **4l** in analytically pure form proved challenging due to partial decomposition during purification. In addition, as indicated in their structures, most ynimines exist as E/Z mixtures, which may give the impression that some compounds are not spectroscopically clean.

Reviewer #3

“Zhu et al. present a new method for synthesizing 2-imido-1,3-dienes from ynimines and carboxylic acids. They further introduce a three-component reaction with ynimines, carboxylic acids, and electron-deficient alkenes to create 1-imido-3,4-trans-disubstituted cyclohex-1-enes via hydroacyloxylation, Mumm rearrangement, and Diels–Alder cycloaddition. An intramolecular version accesses trans-fused tricyclic scaffolds similar to trans-Δ⁹-tetrahydrocannabinol, demonstrating this strategy’s versatility. Chemoselective hydrolysis yields 2-amido-1,3-dienes, which undergo chiral squaramide-catalyzed Diels–Alder reactions for 1-amido-3,4-cis-disubstituted cyclohex-1-enes with high selectivity. The different stereochemical outcomes arise from distinct mechanistic pathways influenced by sterics and electronics. Overall, this work offers strong atom economy and broad utility and is recommended for publication in Nature Communications after minor revisions noted below.”

We thank the reviewer for the positive and encouraging comment and have addressed the points raised below.

Regarding “*Were strongly electron-withdrawing β-aryl groups (e.g., nitro, cyano, CF₃) tested?*” The reaction did occur when ynimine bearing a 4-nitrophenyl group at its beta position. However, we were unable to purify the reaction mixture due to the decomposition of the cycloadduct. We added a sentence in the revised version and it reads (page 4, lines 15-16): “*However, the ynimine bearing a 4-nitrophenyl group at its β position (Ar = 4-NO₂C₆H₄) led to an unstable cycloadduct that decomposed upon purification.*”

Regarding “For substrate **7i** in Figure 3, is there a mixture of regioisomers/diastereomers, and which predominates? How was selectivity determined? (e.g. based on ¹H NMR or GC/HPLC of crude reaction mixtures?)”. This is a typo. Only one diastereomer was detected for **7i**, consistent with the other compounds of this series. This has been corrected in the revised version.

Regarding “Did the study assess 2-imido-1,3-dienes, acrylates, or acrylonitriles as dienophiles in enantioselective Diels–Alder reactions?” We have examined the enantioselective cycloaddition of in situ generated 2-amido-1,3-diene with methyl acrylate. The reaction afforded the cycloadduct in low ee. A sentence was added and it reads (page 7, lines 3-4): “In addition to α,β -unsaturated ketones, methyl acrylate underwent cycloaddition but provided the cycloadduct with low ee.” Since acrylonitrile failed to undergo the cycloaddition in the racemic version, we did not examine its enantioselective version (*cf.*, reply to question 2 of reviewer 2).

Regarding “Can the authors clarify if side products formed or if starting material was recovered, given lower yields in both the enantioselective and the intramolecular Diels–Alder reactions?” All ynimines are fully consumed and the reaction proceeds cleanly. The moderate isolated yield may be attributable to partial decomposition of the starting materials. To clarify this point, we have added the following sentence in the revised manuscript (page 7, lines 15-18): “Although the overall yield appears to be moderate, it should be noted that the conversion of **15** and **5a** to **16** involves at least five individual bond-forming events, including the formation of two C-C, one C-N, one C=O, and two C-H bonds. On this basis, the effective yield per bond formation is comparatively high.”

Regarding “Was telescoping (two consecutive Diels–Alder reactions using ortho-hydroxybenzhydryl alcohols) attempted as in prior work [Angew. Chem. Int. Ed. 62, e202214925 (2023)]?” We thank reviewer for this suggestion. We regret that this experiment was not conducted. In the course of this study, our objective was to explore an application that differ from previously published work, which motivated our investigation of the intramolecular cycloaddition (**15** to **16**, Fig. 6).

Regarding “In Figure 3, replace RCOOH with PhCO₂H since only benzoic acid was used.” In response to Reviewer 1’s suggestion, we have included an additional example using glycolic acid as the reaction partner. Accordingly, the original equation has been retained.

Regarding “The experimental molecular formula for compound **7s** does not match its reported HRMS data.” The reported HRMS data are in fact correct for compound **7s** (it became **7t** in the revised version) However, there was a typo in the molecular formula. The correct formula is C₃₄H₃₇NNaO₄ instead of C₃₃H₃₇NNaO₃. The error results from the copy-and-paste from compound **7r** (became **7s** in the revised version). We apologize for this oversight, which has now been corrected in the revised manuscript.

We thank again referees for their insightful comments and thank you for handling our manuscripts.

Best regards

Yours sincerely,

Prof. Jieping Zhu, PhD

Lausanne, 08-January-2026

Dear Editor,

Please find enclosed our revised manuscript entitled “Ynimines as Versatile Precursors to 2-Imido- and 2-Amido-1,3-Dienes for Stereodivergent Diels–Alder Reactions” (MS N° NCOMMS-25-96860). We thank reviewers for their thoughtful comments. The manuscript has been revised following your advice and reviewers’ suggestions.

Reviewer #1

“This paper describes stereoselective Diels-Alder reactions with ynimines, as precursors of 2-imido- and 2-amido-1,3-dienes. The authors conducted many reactions and are actively investigating them; however, this reviewer has several comments as mentioned below.”

We thank the reviewer for the positive and encouraging comment and have addressed the points raised below.

Regarding “1. *Since the chemical yield of this reaction is low and the ee is not very good, it is not very attractive from the perspective of organic synthetic chemistry.*” The reaction reported in this manuscript constitutes a multiple bond-forming process. While we agree with the reviewer that the overall isolated yields for some examples are moderate, the transformation can nevertheless be regarded as efficient, as it forges multiple bonds in a single operation, specifically two C–C bonds, one C–N bond, one C=O bond, and two C–H bonds. To clarify this point, we have added the following sentence to the revised manuscript (page 7, lines 15–18): *“Although the overall yield appears to be moderate, it should be noted that the conversion of **15** and **5a** to **16** involves at least five individual bond-forming events, including the formation of two C-C, one C-N, one C=O, and two C-H bonds. On this basis, the effective yield per bond formation is comparatively high.”*

Although the enantioselectivity is not perfect, we consider it to be good for this type of transformation. While the attractiveness of a given reaction can be subjective, the synthetic utility of enamides generated from our reaction has been extensively documented (for representative reviews, see references 22–24).

Regarding “2. *Similar structures to 2-amido-1,3-dienes **6a** are commercially available. This reviewer thinks the 1-imido form also can be derived from this, and this reaction can be achieved without **4a**, which must be synthesized separately. Therefore, this synthetic method is not particularly attractive.*” We appreciate the reviewer’s comment and agree that compounds with structures related to **6a** may be commercially available. Nevertheless, despite careful searches, we were unable to identify such examples. When the structure of 2-imido-1,3-diene **6a** was queried in SciFinder, only two entries were returned: a) United States, US2446172 **1948**-08-03 and b) *Chemisches Zentralblatt*, **1948**, 119 Suppl. 2, 1158-1158. Unfortunately, neither source is accessible to us. Moreover, both documents share the same title: 2-

acylamino-buta-1,3-diene, which describes compounds distinct from **6a**, as the latter bears a 2-(N-diacylamino) substituent.

We also acknowledge the reviewer's point: "*1-imido form also can be derived from this, and this reaction can be achieved without 4a*" as the reviewer stated." However, we would like to emphasize two key aspects of the present study. First, the manuscript specifically addresses the synthesis and application of 2-imido- and 2-amido-1,3-dienes, which are considerably more challenging to access than their 1-imido counterparts and remain underexplored. This motivation is clearly stated in the Introduction (page 1, lines 27–31): "*Interestingly, while the cycloaddition reactions of 1-amido-1,3-dienes **1**⁵⁻¹⁹, including catalytic enantioselective variants, have been well documented for the synthesis of allylic amines,^{20,21} research on the 2-amido counterpart **2** has largely lagged behind. This is unfortunate, as the cycloaddition of 2-amido-1,3-dienes **2** affords an enamide function which is poised for further functional group transformations²²⁻²⁴. The lack of progress in this area is likely due to the absence of a convenient and general synthetic method for preparing **2***". Second, the formation of 1-imido-1,3-dienes from ynimines represents a competing side pathway that we intentionally minimized, as discussed on page 5, lines 3–11 and shown in Fig. 4. In this context, the ability to selectively access 2-imido- and 2-amido-1,3-dienes constitutes an important and distinguishing feature of our approach.

Regarding "3. *The asymmetric catalyst is not a new one that was newly developed by the authors.*" While the catalyst is known, the squaramide-catalyzed enantioselective cycloaddition of 2-amido-1,3-dienes reported herein has not been previously described and constitutes a novel contribution.

Regarding "4. *More detailed consideration of the reaction mechanism using calculations, etc. is needed.*" Although DFT calculations could be informative, we believe they are not essential for the scope of this work, which is centered on reaction development.

Regarding "5. *There is no mention of applications of the products (natural products, biologically active substances, etc.)*." We appreciate the reviewer's perspective and acknowledge that applications to natural products or biologically active compounds can be valuable in demonstrating the utility of new methodologies. However, we note that such applications are not always essential for the publication of a new synthetic method. Nevertheless, we would like to emphasize that the methodology developed in this study has indeed been applied to the synthesis of analogues of *trans*- Δ^9 -tetrahydrocannabinol, as described on page 7 in the subsection entitled "Intramolecular Diels–Alder reaction of in situ generated 2-imido-1,3-dienes" and illustrated in Fig. 6.

Reviewer #2

*"1-Amido-cyclohex-1-ene motifs are widely found in the pharmaceutical, natural products and biologically active compounds. This manuscript by Zhu and coworkers reported a facile method for the synthesis of 2-imido-1,3-dienes via the reaction of ynimines with carboxylic acid. Authors found that a three-component reaction of ynimines, carboxylic acids and electron-deficient alkenes enable the efficient synthesis of 1-imido-3,4-trans disubstituted cyclohex-1-enes. Furthermore, the 2-imido diene can be selectively removed to access two distinct 2-amido-1,3-dienes, which undergo chiral squaramide-catalyzed enantioselective Diels-Alder reactions to furnish 1-amido-3,4-cis-disubstituted cyclohex-1-enes with excellent diastereo- and enantioselectivity. The stereochemical divergence is proposed by a concerted *exo*-TS for 2-imido-1,3-dienes and a stepwise pathway for 2-amido-1,3-dienes. In my opinion, publication of the manuscript may be recommended in Nature Communications after a few points were supplemented and revised, which are as follows"*

We thank the reviewer for the positive and encouraging comment and have addressed the points raised below.

Regarding “1. In Page 3 line 24, the Fig. 2c should be Fig. 2b.” Corrected.

Regarding 2. in Fig 3, author investigate different dienophiles partners, such as ethyl vinyl ketone, ethyl acrylate, 4-bromo phenyl acrylate, and 4-chlorophenyl acrylate, I think that acrylonitrile and nitroethylene should be added.” We have examined two dienophiles. However, no cycloaddition products were observed, likely due to the facile polymerization of these highly active species. A sentence was added and it reads (page 4, lines 18-19): “However, no cycloaddition products were observed when acrylonitrile and nitroethylene were used as dienophiles, likely due to the facile polymerization of these highly active species.”

Regarding “3. As for Fig 3, only 7i with dr 11:1, other 7 products were all single trans-figure? If yes, I think that author should mention it in the manuscript.” This was a typo. Only one diastereomer was detected for **7i**, consistent with the other compounds of this series. This has been corrected in the revised version.

Regarding “4. in three-component reaction, can the alky acid afford the desired products?” Glycolic acid was used and the reaction afforded the cycloadduct in good yield but with reduced diastereoselectivity. A sentence was added and it reads (page 4, line 19, and page 5, lines 1-2): “Finally, glycolic acid participated in this three-component reaction to afford cycloadduct **7r** in 55% yield with diminished diastereoselectivity (dr 5:1).”

Regarding “5. Many formatting errors need to be corrected, such as reference 20, 24, 27, 30, 37, etc. letter of all words should be capital.” It has been corrected in the revised version. All other references have been verified carefully and the first letter of each word in the title are capitalized.

Regarding “6. Some NMR spectra are not clean, including impurities or residual solvents (1H NMR-4l, 6a). please provide clean copies.” The ¹H and ¹³C NMR spectra of compound **6a** recorded in CD₃CN exhibit significantly higher resolution than those previously obtained in CD₃Cl. Accordingly, the earlier data (Page S17) and spectra (Page S96-S97) have been replaced with those acquired in CD₃CN. Isolation of compound **4l** in analytically pure form proved challenging due to partial decomposition during purification. In addition, as indicated in their structures, most ynimines exist as E/Z mixtures, which may give the impression that some compounds are not spectroscopically clean.

Reviewer #3

“Zhu et al. present a new method for synthesizing 2-imido-1,3-dienes from ynimines and carboxylic acids. They further introduce a three-component reaction with ynimines, carboxylic acids, and electron-deficient alkenes to create 1-imido-3,4-trans-disubstituted cyclohex-1-enes via hydroacyloxylation, Mumm rearrangement, and Diels–Alder cycloaddition. An intramolecular version accesses trans-fused tricyclic scaffolds similar to trans- Δ^9 -tetrahydrocannabinol, demonstrating this strategy’s versatility. Chemoselective hydrolysis yields 2-amido-1,3-dienes, which undergo chiral squaramide-catalyzed Diels–Alder reactions for 1-amido-3,4-cis-disubstituted cyclohex-1-enes with high selectivity. The different stereochemical outcomes arise from distinct mechanistic pathways influenced by sterics and electronics. Overall, this work offers strong atom economy and broad utility and is recommended for publication in Nature Communications after minor revisions noted below.”

We thank the reviewer for the positive and encouraging comment and have addressed the points raised below.

Regarding “Were strongly electron-withdrawing β -aryl groups (e.g., nitro, cyano, CF₃) tested?” The reaction did occur when ynimine bearing a 4-nitrophenyl group at its beta position. However, we were unable to purify the reaction mixture due to the decomposition of the cycloadduct. We added a sentence in the revised version and it reads (page 4, lines 15-16): “However, the ynimine bearing a 4-nitrophenyl group at its β position (Ar = 4-NO₂C₆H₄) led to an unstable cycloadduct that decomposed upon purification.”

Regarding “For substrate **7i** in Figure 3, is there a mixture of regioisomers/diastereomers, and which predominates? How was selectivity determined? (e.g. based on ¹H NMR or GC/HPLC of crude reaction mixtures?)”. This is a typo. Only one diastereomer was detected for **7i**, consistent with the other compounds of this series. This has been corrected in the revised version.

Regarding “Did the study assess 2-imido-1,3-dienes, acrylates, or acrylonitriles as dienophiles in enantioselective Diels–Alder reactions?” We have examined the enantioselective cycloaddition of in situ generated 2-amido-1,3-diene with methyl acrylate. The reaction afforded the cycloadduct in low ee. A sentence was added and it reads (page 7, lines 3-4): “In addition to α,β -unsaturated ketones, methyl acrylate underwent cycloaddition but provided the cycloadduct with low ee.” Since acrylonitrile failed to undergo the cycloaddition in the racemic version, we did not examine its enantioselective version (*cf.*, reply to question 2 of reviewer 2).

Regarding “Can the authors clarify if side products formed or if starting material was recovered, given lower yields in both the enantioselective and the intramolecular Diels–Alder reactions?” All ynimines are fully consumed and the reaction proceeds cleanly. The moderate isolated yield may be attributable to partial decomposition of the starting materials. To clarify this point, we have added the following sentence in the revised manuscript (page 7, lines 15-18): “Although the overall yield appears to be moderate, it should be noted that the conversion of **15** and **5a** to **16** involves at least five individual bond-forming events, including the formation of two C-C, one C-N, one C=O, and two C-H bonds. On this basis, the effective yield per bond formation is comparatively high.”

Regarding “Was telescoping (two consecutive Diels–Alder reactions using ortho-hydroxybenzhydryl alcohols) attempted as in prior work [Angew. Chem. Int. Ed. 62, e202214925 (2023)]?” We thank reviewer for this suggestion. We regret that this experiment was not conducted. In the course of this study, our objective was to explore an application that differ from previously published work, which motivated our investigation of the intramolecular cycloaddition (**15** to **16**, Fig. 6).

Regarding “In Figure 3, replace RCOOH with PhCO₂H since only benzoic acid was used.” In response to Reviewer 1’s suggestion, we have included an additional example using glycolic acid as the reaction partner. Accordingly, the original equation has been retained.

Regarding “The experimental molecular formula for compound **7s** does not match its reported HRMS data.” The reported HRMS data are in fact correct for compound **7s** (it became **7t** in the revised version) However, there was a typo in the molecular formula. The correct formula is C₃₄H₃₇NNaO₄ instead of C₃₃H₃₇NNaO₃. The error results from the copy-and-paste from compound **7r** (became **7s** in the revised version). We apologize for this oversight, which has now been corrected in the revised manuscript.

We thank again referees for their insightful comments and thank you for handling our manuscripts.

Best regards

Yours sincerely,

Prof. Jieping Zhu, PhD